# The Ophthalmology Surgical Competency Assessment Rubric for Intravitreal Injections (ICO-OSCAR:IVI)

**DOI:** 10.3390/jcm10071476

**Published:** 2021-04-02

**Authors:** Sieh Yean Kiew, Ian Yew San Yeo, Karl C Golnik, María A Muriel-Herrero, Vanesa Fuertes-Barahona, Andrzej Grzybowski

**Affiliations:** 1Singapore National Eye Centre, Singapore 169856, Singapore; kiew.sieh.yean@singhealth.com.sg (S.Y.K.); ian.yeo.y.s@singhealth.com.sg (I.Y.S.Y.); 2Duke-NUS Medical School, National University of Singapore, Academic Clinical Programme for Ophthalmology, Singapore 169856, Singapore; 3Department of Ophthalmology, University of Cincinnati, The Cincinnati Eye Institute, Cincinnati, OH 45242, USA; golnikkarl@gmail.com; 4Hospital Universitario de La Princesa, 28006 Madrid, Spain; maramurielh@gmail.com (M.A.M.-H.); vanesa.fuertes.barahona@gmail.com (V.F.-B.); 5Department of Ophthalmology, University of Warmia and Mazury, 10-561 Olsztyn, Poland; 6Foundation for Ophthalmology Development, Institute for Research in Ophthalmology, 60-836 Poznan, Poland

**Keywords:** assessment, intravitreal injections, rubric, ICO-OSCAR, education

## Abstract

(1) Background: Intravitreal injections (IVIs) are the most commonly performed intraocular procedure worldwide. Ensuring correct learning is essential to avoid complications. Our purpose was to develop an internationally valid tool to assess skill in performing IVIs. (2) Methods: A panel of six content experts designed a rubric for assessing the IVI procedure by using a modified Dreyfus scale of skill acquisition, dividing it into steps and providing objective behavioral descriptors for each level of skill in each category, following the International Council of Ophthalmology (ICO) guidance. The rubric draft was then critically reviewed by 12 international content experts, and their constructive comments were considered for the final rubric. (3) Results: The Ophthalmology Surgical Competency Assessment Rubric for IVI (ICO-OSCAR:IVI) is the proposed tool for assessing healthcare professionals training to perform IVI. (4) Conclusions: The ICO-OSCAR:IVI is the result of the consensus of an international expert panel. The methodology used for its development provides this rubric with face and content validity. It can be used globally to assess healthcare professionals training to perform IVI, as well as the impact of different teaching methods on performance. Further studies are required to establish intra- and inter-rater reliability, as well as the predictive validity of this tool.

## 1. Introduction

Intravitreal injections (IVIs) of air were first described in 1911 by Ohm [1] as an adjunct in retinal-detachment surgery. Once restricted to an intraoperative adjunct, or for intraocular drug delivery in endophthalmitis [2], IVIs have seen logarithmic growth in the past decade, and their expanding use represents one of the most dynamic areas in ophthalmology. Due largely to the introduction of novel drug therapies, expanding indications, and innovation in intravitreal drug-delivery systems and implants, IVIs are now the most commonly performed ophthalmological procedure. Over 4 million IVIs were performed in the U.S. in 2013, rising further to an estimated 5.9 million injections in 2016 [3].

Though considered a simple procedure, it is not exempt from risks and potentially serious complications if not performed properly [4]. Taking into account these safety concerns, ensuring a correct skill acquisition from the very beginning of the learning curve is essential, and there is a need for a standardized tool to aid training and assessment of trainee competence at administering IVIs, which could be used in ophthalmology residency programs worldwide.

The International Council of Ophthalmology (ICO) has been committed to the development of this kind of tool for many years now, providing ophthalmic educators worldwide with the Ophthalmology Surgical Competency Assessment Rubric (ICO-OSCAR). So far, the ICO-OSCAR is available for a number of ophthalmic surgical procedures, such as cataract surgery [5,6], trabeculectomy [7], vitrectomy [8], or strabismus surgery [9], and also for non-intraocular procedures such as laser photocoagulation [10]. To our knowledge, such a tool has not yet been developed or published for IVIs.

Our objective was to develop a standardized, internationally valid tool to both guide the development and assess the progression of a healthcare professional’s competence in performing IVIs. The present study describes our methods of designing an assessment tool for IVIs, and presents the result of an international expert panel consensus for this skill-based rubric.

## 2. Materials and Methods

All procedures during this study were performed in adherence to the Declaration of Helsinki.

A literature review was performed in both the PubMed and Embase databases, using the search terms “intravitreal”, “intravitreal injection”, “assessment”, “evaluation”, and “rubric”, producing no results. An additional search for the development of previously published ICO-OSCARs was also carried out in PubMed, introducing “ICO-OSCAR” and “Ophthalmology Surgical Competency Assessment Rubric” as search terms, and the same methodology was followed for the present study [5,6,7,8,9,10].

This rubric was composed in English, as this is the current internationally accepted lingua franca within the field of ophthalmology. Taking into account expert consensus and published guidelines for international best practices in the administration of IVIs [11,12], the authors (representing faculty from Spain, Poland, Singapore, and the United States) worked together as content experts to create the rubric as follows.

First, all the technical aspects of the procedure were broken down into sequential steps, which are represented in the rubric rows. Cognitive understanding of the indications, patient selection, and counseling were excluded by the authors, as the rubric is intended to assess only procedural competence (in line with previous ICO-OSCAR rubrics). Assessment of clinical knowledge, decision-making or communication skills is beyond the scope of this rubric as well.

Second, columns were assigned different learning stages using a modified Dreyfus model for skill acquisition [13], including the following categories: novice, beginner, advanced beginner, and competent. Expert-level descriptors from the original Dreyfus model were not included in this rubric, as this is not usually achievable during residency or fellowship training. Rating categories were also assigned a numerical value so that an average numerical rating or total score for the tool could be calculated and monitored for improvement.

Each cell in the resulting table was then filled with clear behavioral narrative anchors and stratifying aspects of the skill acquisition expected for each of the mentioned categories, from novice to competent (e.g., familiarity with procedural steps, amount of prompting required, or ability to cope in nonstandard situations). Descriptors were detailed in such a way that could eliminate bias in learner assessment, remove any conjectures of a learner’s completion of steps, and reduce interpretative error of the criteria. Content validity for the rubric was achieved by reaching full consensus among the authors.

With the intention of demonstrating face validity, the rubric was then disseminated online to an expert panel consisting of 10 representatives from Italy (1), Finland (1), Israel (1), the United Kingdom (1), Australia (1), New Zealand (1), Argentina (1), Mexico (1), and the United States (2). This members of this review panel have expertise in IVI training, and were chosen for a broad international representation. Comments by the panel were discussed, documented, and incorporated into the final document.

## 3. Results

The final rubric was given the name “Ophthalmology Surgical Competency Assessment Rubric for Intravitreal Injections” (ICO-OSCAR: IVI), and it is shown in Table A1 in Appendix A.

The IVI procedure was split into a total of 10 steps, which were then classified into 3 groups:(1)Preprocedure considerations (rows 1–6);(2)Injection procedure (rows 7–8);(3)Post-procedure considerations (rows 9–10).

An additional section was included for global indices (rows 11–14) to assess overall procedural smoothness, fluidity, and other important principles such as sharps handling and disposal.

Given the wide variation in technique and clinical settings in which IVIs are administered worldwide, where steps were not felt to be mandatory or compulsory, notes were included within the final rubric to allow for variations with regards to patient positioning, use of speculum or other methods of eyelid retraction to prevent contamination of injection site, and choice of local anesthetic agent. As an example, recent European consensus recommendations [11] concluded that the use of gloves (either sterile or nonsterile) when administering IVIs is appropriate and consistent with modern clinical practice; however, given the lack of data in the published literature for or against the use of gloves, this has been left optional in the rubric.

Several minor adverse events that were initially included in the rubric, including perception of pain by the patient and subconjunctival hemorrhage post-procedure, were also omitted from the final document, as expert consensus was that these events may occur even despite properly performed technique.

## 4. Discussion

IVIs are currently the most commonly performed ophthalmological procedure worldwide, yet teaching methods and trainee experience varies greatly within training programs internationally. Eighty-seven percent of respondents in a 2009 survey of U.S. residency program faculty identified IVI as a procedure in which graduating residents would be expected to be competent. However, the same survey found that based on case-log data, only 14% of residents graduating in 2009 had performed more than 20 IVIs during their residency [14]. Residency programs in the United States are now required to show in aggregate that residents have performed a specified “minimum” number of IVIs, and numbers have certainly increased since then. As a sample, Accreditation Council for Graduate Medical Education (ACGME) case-log data shows that the average number of intravitreal injections (with or without accompanying aqueous or vitreous tap) has tripled from 31.3 procedures per resident in 2009 to 93.1 procedures per resident in 2015 [15], but there are as yet no standard requirements or measures to assess how well the resident performed, or to certify that residents are clinically competent in the procedure.

A review of the literature found no universal or standardized tools for teaching and assessing the IVI technique. The authors believe that there is a need for such a tool, to ensure that the technique is properly learned from the very beginning of the learning curve of a trainee, given that it is not exempt from potentially serious complications that can easily be avoided by using a correct technique [4].

This rubric does not pretend to serve as a clinical-practice guideline. In fact, the main difficulty in its development process was to take into account so many different ways in which the technique is performed in a variety of regions and countries, particularly with regard to surgeon and surgical-field preparation. It is the understanding of this panel that variations in practice are acceptable so long as sterility and safety are not compromised, hence the presence of “optional” items within the rubric. The aim of the proposed ICO-OSCAR:IVI is to provide the ophthalmic educator community around the world with a helpful tool to objectively assess whether their trainees are performing the IVI technique properly, taking into account local variations in practice specific to their setting.

Teaching and training in any busy setting is time-consuming, but the aim of training and subsequent assessment is to produce competence, after which the trainee graduates and no longer needs to be assessed continually for every case, so it is the authors’ view that the investment in time taken to train, assess, and debrief is amply repaid by creating a pool of skilled and adequately credentialed people who can subsequently carry out IVIs safely and efficiently. Filling out the ICO-OSCAR:IVI and providing feedback to the trainee is expected to take about 10 min per case. It is advisable that both the instructor and trainee have access to the rubric´s content before performing the procedure, as this will increase the fluidity of the evaluation process. In pilots using this rubric at the Singapore National Eye Centre (unpublished data), where the assessor has filled the form while observing the procedure carried out, the form-filling does not add any significant time to the length of the procedure itself. Subsequent debriefing with the trainee of course takes time, but is an important component of the learning cycle, providing feedback for growth and improvement.

The ICO-OSCAR: IVI should be used while an individual is gaining competence and not afterward. The number of times it should be used depends on an individual’s ability to become competent. We suggest that once an individual demonstrates competency on at least five successive OSCARs on two different sessions, there is no need to continue its use, but this decision should be at the discretion of each instructor.

Similar OSCARs have been developed and disseminated by the ICO in several languages for other common ophthalmic surgical procedures, such as phacoemulsification surgery [5], extracapsular cataract extraction [6], strabismus surgery [9], vitrectomy [8], and laser procedures such as panretinal photocoagulation [10].

Some ophthalmic educators may think that this rubric lacks relevant items related to transversal skills. These were initially considered by the authors, but were excluded from the draft to maintain coherence with the ICO-OSCAR standard, which is focused on the assessment of procedural skills, whereas transversal skills such as interpersonal and professional behavior, communication with patients and other members of the medical team, professionalism, and system-based practice can be evaluated with the ICO 360° Assessment Tool [16].

In keeping with the other OSCARs (such as the ICO-OSCAR: phaco), a limitation of this rubric is that it evaluates only procedural competence, and does not address clinical decision-making (patient selection, drug choice) nor patient education or counseling, all of which remain important aspects of clinical competence in ophthalmology surrounding the IVI procedure. Apart from recognition of the most serious complications (visual loss due to central retinal artery occlusion and/or significant rise in intraocular pressure), this rubric does not address the management of complications either. These aspects of competence (clinical decision-making, patient communication and education, recognition and management of post-procedure complications) should be addressed separately, and the ICO-OSCAR: IVI tool used as an adjunct to assess technique as part of the learner’s overall evaluation. Additionally, it is important to note that this rubric is specific to intravitreal drug injections only, and does not include the delivery of sustained-release intravitreal drugs (such as the Ozurdex TM or Retisert TM sustained-release implants). The delivery of sustained-release intravitreal drugs is different enough from a standard intravitreal injection that the authors and international expert panel are of the opinion that these should not be covered by the same assessment scale as that for IVIs.

This ICO-OSCAR:IVI rubric has been developed and validated in English, as it is the most frequently used language in the ophthalmology community worldwide. We believe that the terminology used in the rubric is easy to understand for medium-level, non-native English speakers. In fact, three authors and five members of the international validation panel do not have English as the official language in their countries. Once published, this OSCAR can be translated into different languages as needed to broaden its usefulness, as has been done with other ICO-OSCARs.

Further studies are also required to evaluate intra- and inter-rater reliability, as well as the predictive value of this tool.

In conclusion, the standardized rubric presented in the current study can serve as a useful tool for teaching, training, and evaluating trainees´ performance before being credentialed for administration of IVIs. It could also be used to periodically reaccredit all injectors to ensure safety is not compromised. Having been compiled by an international panel of authors with specialty expertise and subsequently reviewed by an international panel of experts, our rubric demonstrates both face and content validity, and can be used globally.

## Data Availability

Not applicable.

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
