# Peer review of "The Ophthalmology Surgical Competency Assessment Rubric for Intravitreal Injections (ICO-OSCAR:IVI)"

_jcm, 2021, doi:10.3390/jcm10071476_

Round 1
Reviewer 1 Report
The authors have submitted a well written manuscript, and propose an interesting and indeed a very relevant tool to assess intravitreal injections (IVI) among ophthalmic trainees and surgeons, addressing an important issue which is to ensure uniformity, high standard and safety in these procedures. The ICO-OSCAR:IVI scale proposed by the authors is detailed and clear.
However, there are certain major aspects that authors should address and change in the manuscript. The authors may find enclosed in this revision a PDF of the manuscript file with highlights and comments; minor edition aspects (including grammar and rephrasing aspects) are presented in the revised manuscript instead of discussed in this letter.
- While it is true that IVI are performed by non-physicians in some centres, I consider this to be inadequate (despite acknowledging the advantages of this strategy). If the authors believe this to be an adequate strategy, they should also acknowledge that an ICO-OSCAR should not be applied to non-ophthalmologists. That is to say, ICO-OSCAR:IVI should not be a tool to assess non-physicians. Otherwise, the scale evaluators and descriptors must not include the word “surgeon”. Therefore, all mentions to non-physician healthcare professionals should be removed from the manuscript.
- Materials and Methods line 68: provide the methods used for literature search of assessment tools for administering IVI (keywords, databases used, etc).
- Materials and Methods: the language used in the ICO-OSCAR:IVI scale must be specified.
- I am not certain that ICO-OSCAR:IVI could be validated internationally in countries in which English is not the native language, unless the scale in the respective official language (for instance, in this study group, while English in an official language in both the US and Singapore, this scale should be validated in Spanish (Castellano) and in Polish).
- While this appears to be a valuable tool for assessing IVI procedures, how much time do the authors consider this table takes to fill? In high-volume IVI clinics/centres, this scale would be time consuming for the evaluation of the performance; that is to say, if the optimal case duration would be ≤ 10 minutes (evaluator 14), having to individually fill this scale for each IVI (including signatures) would cumulatively lead to increased durations for each case.
- How many IVI procedure evaluations using the ICO-OSCAR:IVI scale do the authors consider to be necessary to ensure the optimization, uniformity and high quality of the IVI performed by ophthalmic trainees and specialists? Do the authors consider that all procedures should be evaluated using this scale every time? Is there a cut-off after which surgeons do not require performing these assessments? And is there a minimum amount of IVI per unit of time (week, month, year) that the authors consider should imply repeating the ICO-OSCAR:IVI evaluation? These aspects should be commented in the Discussion section.
- Limitations of the current study should be mentioned in one or two sentences in the Discussion.
- Do the authors believe that sustained-release intravitreal drugs could be assessed using ICO-OSCAR:IVI as well? If so, perhaps the evaluator 6 “preparation of drug” should include a clause for these type of devices.
Author Response
Reply to reviewers
We would like to thank reviewers for their valuable comments that we included within the revised version of the manuscript.
Reviewer 1:
The authors have submitted a well written manuscript, and propose an interesting and indeed a very relevant tool to assess intravitreal injections (IVI) among ophthalmic trainees and surgeons, addressing an important issue which is to ensure uniformity, high standard and safety in these procedures. The ICO-OSCAR:IVI scale proposed by the authors is detailed and clear.
However, there are certain major aspects that authors should address and change in the manuscript. The authors may find enclosed in this revision a PDF of the manuscript file with highlights and comments; minor edition aspects (including grammar and rephrasing aspects) are presented in the revised manuscript instead of discussed in this letter.
Thank you for your comments and help in improving the edition of our manuscript. In response to your comments:
- While it is true that IVI are performed by non-physicians in some centres, I consider this to be inadequate (despite acknowledging the advantages of this strategy). If the authors believe this to be an adequate strategy, they should also acknowledge that an ICO-OSCAR should not be applied to non-ophthalmologists. That is to say, ICO-OSCAR:IVI should not be a tool to assess non-physicians. Otherwise, the scale evaluators and descriptors must not include the word “surgeon”. Therefore, all mentions to non-physician healthcare professionals should be removed from the manuscript.
We accept that the administration of intravitreal injections by non-physician healthcare professionals is controversial and not universally accepted; as such we have revised as suggested and removed all mentions to non-physician healthcare professionals within this manuscript as well as in References.
- Materials and Methods line 68: provide the methods used for literature search of assessment tools for administering IVI (keywords, databases used, etc).
We have included specific information about consulted databases and keywords used in the literature search.
- Materials and Methods: the language used in the ICO-OSCAR:IVI scale must be specified.
This rubric was developed in English, as the current internationally-accepted lingua franca in Ophthalmology. We have included a statement to this effect in the Methods section.
- I am not certain that ICO-OSCAR:IVI could be validated internationally in countries in which English is not the native language, unless the scale in the respective official language (for instance, in this study group, while English in an official language in both the US and Singapore, this scale should be validated in Spanish (Castellano) and in Polish).
The original rubric has been developed and validated in English as it is the most frequently used language in the Ophthalmology community worldwide, and easy to understand for "medium-level" non-native English speakers. Once the original version is published, it can be translated as other ICO-OSCARs into as many different languages as needed to broaden its usefulness. We have added a comment in the Discussion Section.
- While this appears to be a valuable tool for assessing IVI procedures, how much time do the authors consider this table takes to fill? In high-volume IVI clinics/centres, this scale would be time consuming for the evaluation of the performance; that is to say, if the optimal case duration would be ≤ 10 minutes (evaluator 14), having to individually fill this scale for each IVI (including signatures) would cumulatively lead to increased durations for each case.
Taking into account other reviewers’ comments, case duration is not only dependent on the surgeon´s technical skills, so we have decided to eliminate the specified times from the rubric´s 14th step.
Apart from this consideration, we are aware that teaching and training in any busy setting are time-consuming, but the aim of training and subsequent assessment is to produce competence, after which the trainee graduates and no longer needs to be assessed continually for every case, so it is the authors’ view that the investment in the time taken to train, assess and debrief is amply repaid by creating a pool of skilled and adequately credentialled people who can subsequently carry out IVI safely and efficiently.
In pilots using this rubric in Singapore National Eye Centre, where the assessor has filled the form while observing the procedure carried out, the form-filling does not add any significant time to the length of the procedure itself. Subsequent debrief with the trainee of course takes time, but is an important component of the learning cycle, providing feedback for growth and improvement.
The time expected for filling the OSCAR and providing feedback is about 10 minutes per case. This can be accomplished by allowing both the instructor and trainee to have access to the rubric´s content before the procedure, increasing the fluidity of the evaluation process.
We have included a comment in the Discussion.
- How many IVI procedure evaluations using the ICO-OSCAR:IVI scale do the authors consider to be necessary to ensure the optimization, uniformity and high quality of the IVI performed by ophthalmic trainees and specialists? Do the authors consider that all procedures should be evaluated using this scale every time? Is there a cut-off after which surgeons do not require performing these assessments? And is there a minimum amount of IVI per unit of time (week, month, year) that the authors consider should imply repeating the ICO-OSCAR:IVI evaluation? These aspects should be commented in the Discussion section.
The OSCAR should be used while an individual is gaining competence and not afterwards. The number of times it should be used depends on an individual’s ability to become competent. Therefore, there is no set number of times. We think that once an individual demonstrates competency on at least 3 successive OSCARs there is no need to continue its use. We have added this in the Discussion.
- Limitations of the current study should be mentioned in one or two sentences in the Discussion.
Limitations of the current rubric (that it assesses only procedural competency without covering other important aspects such as clinical decision-making, patient communication and education, and post-procedure care) have been more clearly described in the Discussion section on page 9.
- Do the authors believe that sustained-release intravitreal drugs could be assessed using ICO-OSCAR:IVI as well? If so, perhaps the evaluator 6 “preparation of drug” should include a clause for these type of devices.
We considered this but ultimately felt that the delivery of sustained-release intravitreal drugs is different enough from a standard intravitreal injection that this should not be covered by the same assessment scale.
Andrzej Grzybowski, MD, PhD

Reviewer 2 Report
This manuscript presents the consensus of a multinational committee of expert intravitreal injection (IVI) teachers. It a grading scheme for ophthalmology residents and others who are learning to perform IVI, breaking down the process into easily understandable categories and defining the grades in each category. The paper is well-written and draws on diverse worldwide opinion. However, it does not offer any new scientific information.
Ophthalmology Surgical Competency Assessment Rubric (OSCAR) is a series of manuscripts (based on expert panel consensus) that define core competencies in various ophthalmology surgical procedures. Previous OSCARs include phaco, extracap, strabismus surgery, vitrectomy, and PRP laser. The proposed OSCAR relates to IVI, which is a feat in itself because IVI is widely performed in many different variations. What is very new about this OSCAR is that it proposes to judge competency for ophthalmology residents and fellows, AND also non-physician practitioners (e.g., nurses). It is a major assumption and sure to be controversial depending on the country or region of the reader. The paper implies that a good OSCAR:IVI score signifies competency that is equivalent, whether the student is an ophthalmology resident, a retina fellow, a nurse in the UK, or an optometrist practicing independently in Oklahoma. This is a discussion beyond the scope of this manuscript. A better approach would be to remove the topic of IVI by non-physicians and address that topic separately. This manuscript should focus on the rubric as it applies to ophthalmology residents and fellows (in line with other OSCARs, and also in line with most of the paper as currently written). Then, if any facility wishes to apply the rubric to non-physicians in their program, they are free to do so.
SPECIFIC RECOMMENDATIONS
- p. 2 ln. 44-63. Regarding the controversial topic mentioned above, I recommend simply acknowledging, "Furthermore, IVI is traditionally administered by ophthalmologists, and less commonly by nurses (with oversight by ophthalmologists) in some countries." Delete 46-47 "and the wider range of healthcare providers who are being trained to perform them." Delete line 50 "not only," and line 51 ", but also other professionals implicated." In line 60, change "healthcare professional's" to "ophthalmologist's." Also, consider a similar adjustment on page 9 line 62-63. If instead the intention of the committee is to argue that OSCAR:IVI competency can be graded the same for non-physicians and ophthalmologists alike, then this concept needs significant expansion and literature support in the introduction and also discussion regarding what the committee recommends for scope of practice, surgical privileges, and oversight for various healthcare professionals.
2. Table 1 section 5: "Aqueous Chlorhexidine" might be less confusion if you instead say, "Topical Chlorhexidine Solution" so that no one thinks this is intracameral chlorhexidine injection into the aqueous (anterior chamber).
3. Table 1 section 8: displacement of the conjunctiva could be optional.
4. Table 1 section 8 or 12: immobilization and/or control of the head, the lids, and the globe is key to minimizing endophthalmitis risk when the patient inevitably flinches or moves involuntarily or blinks more powerfully than the speculum can handle during an injection. A firm/confident/experienced proceduralist can anticipate and control these movements and minimize the risk of the needle touching the lashes or penetrating the globe twice. A timid proceduralist who is afraid to touch the eye will not be ready to mitigate this risk. I recommend this important safety consideration be included in section 8 or 12.
5. page 8 line 33: It is hard to believe that US residents complete 90+ vitreous tap/injections. Perhaps you mean that that US residents complete 90+ intravitreal injections (with or without vitreous/aqueous tap)?
6. Table 1 section 14: putting specific times (eg, 10 minutes) may not be reasonable, depending on the circumstances. If for example there are no technicians or scribes in the resident clinic to aid with consent, prep, anesthesia, medication, instructions, documentation, then a very competent resident might take more than 10 minutes. Conversely, an inexperienced resident who is supported by 1 or more technicians and scribe may finish more quickly even if they have less competency.
7. Page 2 line 80-81, and page 9 line 54-60. A limitation of the OSCAR:IVI grading system is that it does not include any background knowledge, medical decision making, or post-procedure care in the assessment. These are critical to patient care, beyond the procedural competency. Ophthalmology residency and fellowship programs will need to supplement the OSCAR:IVI to get a full picture of the trainee's competency. The manuscript acknowledges that clinical knowledge, medical decision making, and transversal skills are outside of the scope of the OSCAR (for better or for worse), but the manuscript should also acknowledge that post-procedure care, including triage and management of complications are also not addressed.
Author Response
Reply to reviewers
We would like to thank reviewers for their valuable comments that we included within the revised version of the manuscript.
Reviewer 2:
This manuscript presents the consensus of a multinational committee of expert intravitreal injection (IVI) teachers. It a grading scheme for ophthalmology residents and others who are learning to perform IVI, breaking down the process into easily understandable categories and defining the grades in each category. The paper is well-written and draws on diverse worldwide opinion. However, it does not offer any new scientific information.
Ophthalmology Surgical Competency Assessment Rubric (OSCAR) is a series of manuscripts (based on expert panel consensus) that define core competencies in various ophthalmology surgical procedures. Previous OSCARs include phaco, extracap, strabismus surgery, vitrectomy, and PRP laser. The proposed OSCAR relates to IVI, which is a feat in itself because IVI is widely performed in many different variations. What is very new about this OSCAR is that it proposes to judge competency for ophthalmology residents and fellows, AND also non-physician practitioners (e.g., nurses). It is a major assumption and sure to be controversial depending on the country or region of the reader. The paper implies that a good OSCAR:IVI score signifies competency that is equivalent, whether the student is an ophthalmology resident, a retina fellow, a nurse in the UK, or an optometrist practicing independently in Oklahoma. This is a discussion beyond the scope of this manuscript. A better approach would be to remove the topic of IVI by non-physicians and address that topic separately. This manuscript should focus on the rubric as it applies to ophthalmology residents and fellows (in line with other OSCARs, and also in line with most of the paper as currently written). Then, if any facility wishes to apply the rubric to non-physicians in their program, they are free to do so.
Thank you for your comments. We accept that the administration of intravitreal injections by non-physician healthcare professionals is controversial and not universally accepted; as such we have revised as suggested and removed all mentions to non-physician healthcare professionals within this manuscript as well as in References.
SPECIFIC RECOMMENDATIONS
- p. 2 ln. 44-63. Regarding the controversial topic mentioned above, I recommend simply acknowledging, "Furthermore, IVI is traditionally administered by ophthalmologists, and less commonly by nurses (with oversight by ophthalmologists) in some countries." Delete 46-47 "and the wider range of healthcare providers who are being trained to perform them." Delete line 50 "not only," and line 51 ", but also other professionals implicated." In line 60, change "healthcare professional's" to "ophthalmologist's." Also, consider a similar adjustment on page 9 line 62-63. If instead the intention of the committee is to argue that OSCAR:IVI competency can be graded the same for non-physicians and ophthalmologists alike, then this concept needs significant expansion and literature support in the introduction and also discussion regarding what the committee recommends for scope of practice, surgical privileges, and oversight for various healthcare professionals.
We have revised to remove all mentions to non-physician healthcare professionals within this manuscript, as suggested.
- Table 1 section 5: "Aqueous Chlorhexidine" might be less confusion if you instead say, "Topical Chlorhexidine Solution" so that no one thinks this is intracameral chlorhexidine injection into the aqueous (anterior chamber).
We have changed to “topical chlorhexidine solution” in order to avoid confusion as suggested.
- Table 1 section 8: displacement of the conjunctiva could be optional.
We discussed this and concluded that conjunctiva displacement is a relevant safety issue even though many people do not perform it routinely, so we have decided to keep it as it is in the rubric.
- Table 1 section 8 or 12: immobilization and/or control of the head, the lids, and the globe is key to minimizing endophthalmitis risk when the patient inevitably flinches or moves involuntarily or blinks more powerfully than the speculum can handle during an injection. A firm/confident/experienced proceduralist can anticipate and control these movements and minimize the risk of the needle touching the lashes or penetrating the globe twice. A timid proceduralist who is afraid to touch the eye will not be ready to mitigate this risk. I recommend this important safety consideration be included in section 8 or 12.
We have revised to include management of head/globe movement under risk to tissues in section 12.
- page 8 line 33: It is hard to believe that US residents complete 90+ vitreous tap/injections. Perhaps you mean that that US residents complete 90+ intravitreal injections (with or without vitreous/aqueous tap)?
This refers to the ACGME procedure, coded as vitreous tap/injection where IVIs are logged using the same code as a vitreous tap. Nevertheless, we have revised to clarify, as “intravitreal injection, with or without aqueous or vitreous tap”.
- Table 1 section 14: putting specific times (eg, 10 minutes) may not be reasonable, depending on the circumstances. If for example there are no technicians or scribes in the resident clinic to aid with consent, prep, anesthesia, medication, instructions, documentation, then a very competent resident might take more than 10 minutes. Conversely, an inexperienced resident who is supported by 1 or more technicians and scribe may finish more quickly even if they have less competency.
We agree that case duration is not only related to the ophthalmologist´s technical skill, so we have eliminated the times from the rubric as suggested.
- Page 2 line 80-81, and page 9 line 54-60. A limitation of the OSCAR:IVI grading system is that it does not include any background knowledge, medical decision making, or post-procedure care in the assessment. These are critical to patient care, beyond the procedural competency. Ophthalmology residency and fellowship programs will need to supplement the OSCAR:IVI to get a full picture of the trainee's competency. The manuscript acknowledges that clinical knowledge, medical decision making, and transversal skills are outside of the scope of the OSCAR (for better or for worse), but the manuscript should also acknowledge that post-procedure care, including triage and management of complications are also not addressed.
We have revised to include post-procedural care in the discussion on the limitations of this rubric on page 9.
Andrzej Grzybowski, MD, PhD

Round 2
Reviewer 1 Report
The authors have provided concise, coherent answers to the reviewers' comments. However, some points of the authors' replies raised some questions that should be addressed:
1- "In pilots using this rubric in Singapore National Eye Centre, where the assessor has filled the form while observing the procedure carried out, the form-filling does not add any significant time to the length of the procedure itself. Subsequent debrief with the trainee of course takes time, but is an important component of the learning cycle, providing feedback for growth and improvement."
is this data published or personal communication or unpublished data?
although controversial, this statement seems interesting enough to be mentioned in the discussion section of the manuscript.
2- The main goal of developing this OSCAR, according to the authors, is to ensure high-quality, uniformity in the procedure; therefore, it stands to reason that three successful OSCARs in one session is likely a low number, especially in a procedure with such a high-volume.
Perhaps this number should be higher. Scoring five consecutive OSCARS successfully in two or three different sessions on different days/weeks would likely lead to significant improvement and standardization of the assessment and, eventually, of the IVI procedures.
3- The authors answered the question regarding sustained-release intravitreal drugs in a consistent manner, and I fully agree with the authors (which is why I asked the question). This is an important differentiation that deserves a sentence mentioning it in the discussion section of the manuscript
4- Minor typing error:
As a sample, Accreditation Council for 31 Graduate Medical Education (ACGME) case log data shows that average number of 32 intravitreal injections (with or withoug accompanying aqueous or vitreous tap) have 33 tripled from 31.3 procedures per resident in 2009 to 93.1 procedures per resident in 2015 34 [15], - correct typing error to "without"
Author Response
Reply to Reviewer 1
We would like to thank the reviewer for her/his valuable comments that we included within the revised version of the manuscript. Below the point-by-point response is presented
The authors have provided concise, coherent answers to the reviewers' comments. However, some points of the authors' replies raised some questions that should be addressed:
1- "In pilots using this rubric in Singapore National Eye Centre, where the assessor has filled the form while observing the procedure carried out, the form-filling does not add any significant time to the length of the procedure itself. Subsequent debrief with the trainee of course takes time, but is an important component of the learning cycle, providing feedback for growth and improvement."
is this data published or personal communication or unpublished data?
although controversial, this statement seems interesting enough to be mentioned in the discussion section of the manuscript.
Thank you for your comments. This is unpublished data from Singapore National Eye Centre, where this rubric has been trialled for ease of use while training residents and non-physician personnel to perform intravitreal injections. Thus far we have used it as a training tool to provide feedback, rather than an assessment tool, pending validation studies using this rubric. We have included this sentence in the discussion as suggested.
2- The main goal of developing this OSCAR, according to the authors, is to ensure high-quality, uniformity in the procedure; therefore, it stands to reason that three successful OSCARs in one session is likely a low number, especially in a procedure with such a high-volume.
Perhaps this number should be higher. Scoring five consecutive OSCARS successfully in two or three different sessions on different days/weeks would likely lead to significant improvement and standardization of the assessment and, eventually, of the IVI procedures.
After discussion, we agree with this and have amended accordingly to 5 successful OSCARs on two different sessions.
3- The authors answered the question regarding sustained-release intravitreal drugs in a consistent manner, and I fully agree with the authors (which is why I asked the question). This is an important differentiation that deserves a sentence mentioning it in the discussion section of the manuscript
Thank you. We have added a sentence specifying that this rubric does not cover the delivery of sustained-release intravitreal drugs and the rationale for this in the discussion section.
4- Minor typing error:
As a sample, Accreditation Council for 31 Graduate Medical Education (ACGME) case log data shows that the average number of 32 intravitreal injections (with or without accompanying aqueous or vitreous tap) have 33 tripled from 31.3 procedures per resident in 2009 to 93.1 procedures per resident in 2015 34 [15], - correct typing error to "without"
Thank you for catching this spelling error! Have amended this accordingly.

Reviewer 2 Report
Nice edits. Thank you for your consideration of the review.
Author Response
Thank you.